# Automating Reproducibility in
# Medical Imaging with Deep Learning

**Attila Simkó**[1]  [iD]                                        ATTILA.SIMKO@UMU.SE

[1] *Department of Diagnostics and Intervention, Umeå University, Umeå, Sweden*

**Editors:** Accepted for publication at MIDL 2025

## Abstract

Reproducibility remains a critical challenge in deep learning for medical imaging, limiting the reliability and clinical adoption of published research. An automated framework is presented to assess key reproducibility factors — dependencies, training/evaluation code, weights, documentation and licensing — by analyzing GitHub repositories. Validated on manually annotated MIDL 2024 submissions, the system achieves 66.8%-96.9% accuracy across criteria. Applied to 3,682 papers from MIDL, MICCAI, Nature, and arXiv reveals widespread gaps, particularly in sharing model weights and documentation. This approach enables scalable, objective reproducibility assessments and lays the groundwork for integration into peer review workflows. The source code and a live demo is available online[1].

**Keywords:** Reproducibility, Computational Reproducibility, Automation, MIDL

## 1. Introduction

Reproducibility remains a significant challenge in deep learning research—particularly in medical imaging, where robust, verifiable results are essential for clinical adoption. Despite growing awareness (Varoquaux and Cheplygina, 2022) and the introduction of reproducibility guidelines across conferences and journals, many studies continue to omit critical components such as accessible code, trained models, and sufficient documentation (Simkó et al., 2024).

Manual reproducibility assessments——while insightful——are time-consuming, subjective, and difficult to scale. To address these limitations, this work introduces an automated framework designed to evaluate key reproducibility elements systematically.

## 2. Methodology

The automated framework was developed using manually annotated MIDL submissions from 2018 to 2023 as ground truth. Six reproducibility criteria were targeted: dependency declaration, training and evaluation code, availability of pre-trained weights, repository documentation, and licensing. These criteria align well with existing reproducibility checklists adopted by major conferences and journals. To ensure robustness and scalability, the system was implemented in Python using lightweight rule-based methods such as regular expressions and pattern matching.

---

1. https://huggingface.co/spaces/attilasimko/reproduce

Validation was performed on a separate set of submissions from MIDL 2024, which were not used during system development.

The validated framework was then applied to a large-scale dataset of 3682 papers drawn from four major sources: **MIDL** (All 117 accepted submissions from 2024), **MICCAI** (All 1303 submissions from 2023 and 2024), **Nature Communications Medicine** (883 papers filtered using the keywords: "deep learning" or "AI" in the abstract), and **arXiv** (1379 papers published between 2018 and 2023 filtered using the keywords: "deep learning" and "medical imaging" in the abstract).

## 3. Results

The validation results are collected in Table 1. The method shows particularly strong performance on structured repository elements such as training code (84.2%) dependencies (79.1%) and licensing (96.9%). Documentation remained the most challenging criterion due to its unstructured nature and variation in quality.

Table 1: The accuracy of the automated evaluation system compared to manually collected ground truth using MIDL submissions from 2024.

|  | Dependencies | Training | Evaluation | Weights | Documentation | License |
|---|---|---|---|---|---|---|
| MIDL | 79.1% | 84.2% | 79.5% | 78.0% | 66.8% | 96.9% |

Table 2 contains the results of the automated evaluation of the large-scale dataset. Repository availability varied significantly, with only 31–67% of papers linking to public code.

Table 2: Results of the automated evaluation of 3682 papers.

|  | MICCAI | MIDL | Nature | arXiv |
|---|---|---|---|---|
| Public repositories (%) | 66.84 | 56.88 | 31.37 | 30.96 |
| Num. of public repositories | 871 | 310 | 277 | 427 |
| Dependencies (%) | 68.4 | 73.7 | 55.1 | 58.7 |
| Training (%) | 91.7 | 91.9 | 70.6 | 88.8 |
| Evaluation (%) | 89.5 | 89.6 | 68.4 | 83.8 |
| Weights (%) | 29.7 | 31.3 | 25.0 | 34.7 |
| Documentation (%) | 59.8 | 54.4 | 40.4 | 50.2 |
| Licensing (%) | 48.0 | 57.1 | 64.0 | 70.7 |
| Reproducibility score | $3.872 \pm 1.393$ | $3.981 \pm 1.393$ | $3.235 \pm 1.84$ | $3.869 \pm 1.498$ |

## 4. Discussion

While the automated evaluations do not perfectly match manual annotations (e.g., 66.8% accuracy for documentation), they yield reproducibility scores that are close to ground

truth (3.981 vs. 3.629 on MIDL 2024 submissions). This suggests that the system is reliable enough for scalable assessments.

A significant obstacle remains code availability. Of the $1,885$ collected repository links, a staggering 592 are broken——either pointing to deleted or private repositories. This compromises reproducibility entirely. As a recommendation, hosting code under organizational accounts would to ensure long-term access.

Documentation remains the weakest reproducibility factor across all venues, with as few as 40% of repositories providing adequate guidance. Current binary evaluations ("Yes" vs. "No") oversimplify the nuance needed here. This will be re-designed in future iterations.

License files are easy to detect because they follow a consistent format and location on GitHub. Ideally, similar conventions for the other criteria could greatly improve both human and automated reproducibility checks.

Interestingly, licensing results challenge assumptions about academic rigor. Nature explicitly requires code to include a license, yet only 64% of its repositories complied. For MICCAI, despite its reputation, just 48% of its repositories included a license, lower than arXiv's 70.7%. This suggests that reputation and guidelines alone do not ensure reproducibility without clear incentives.

## 5. Conclusions

An automated framework is introduced for assessing reproducibility in deep learning research, with a focus on medical imaging. By applying the framework to a large-scale dataset, it highlighted shortcomings in reproducibility across major publication venues, emphasizing the need for standardized guidelines and improved compliance.

Despite the official guidelines of MICCAI[2], MIDL[3] and Nature[4], their reproducibility standards are challenged by papers collected from arXiv. The presented results argue for integrating automated assessments into submission platforms like OpenReview. Providing real-time feedback during the submission process could help authors resolve missing elements before peer review.

As a short-term solution, a publicly available tool[5] is released that demonstrates the current system. It will remain accessible during the transition to a broader framework capable of evaluating full manuscripts against flexible, customizable criteria. This future version will include deeper assessments of citation impact, data availability, and unclear manuscript details (Ghanbari Azar et al., 2025). The long-term goal of this framework is to make automated evaluations a standard part of the scientific publishing process.

### Acknowledgments

Thank you to all the authors who publish well-maintained code—and to those who don't: this project wouldn't exist without you. This research was supported by grants from Cancer Research Foundation in Northern Sweden (AMP 25-1211).

---

2. https://conferences.miccai.org/2025/en/PAPER-SUBMISSION-GUIDELINES.html#reproducibleresearch

3. https://www.midl.io/reproducibility

4. https://www.nature.com/nature-portfolio/editorial-policies/reporting-standards#availability-of-computer-code

5. https://huggingface.co/spaces/attilasimko/reproduce

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
