# OpenReview forum: "Automating Reproducibility in Medical Imaging with Deep Learning"
_MIDL.io/2025/Short_Papers — MIDL 2025 - Short Papers_

### Official Review · Reviewer_KdRz · 2025-04-29

**Rating:** 4
**Confidence:** 3

**Summary:**

This paper presents a preliminary version of an automated reproducibility assessment framework for medical imaging deep learning research. The authors train the framework on MIDL submissions from 2018 to 2023 using manually annotated labels, and validate it on MIDL 2024 submissions with manually collected ground truth, as well as a wide range of external test datasets without ground truth.

**Strengths:**

•	The authors address the important but often overlooked problem of reproducibility evaluation for prestigious conferences and journals. Preliminary results demonstrate the potential of the proposed framework using publicly available tools.

**Weaknesses:**

•	The test accuracies reported in Table 1 are still relatively low, as discussed in the paper, which may limit the broader applicability of the framework.

---

### Decision · Program_Chairs · 2025-05-01

Accept